# Antarctic krill sequester similar amounts of carbon to key coastal blue carbon habitats

E. L. Cavan [1] ✉, N. Mackay [2], S. L. Hill [3], A. Atkinson [4], A. Belcher [3,5] & A. Visser [6]

The carbon sequestration potential of open-ocean pelagic ecosystems is vastly under-reported compared to coastal vegetation 'blue carbon' systems. Here we show that just a single pelagic harvested species, Antarctic krill, sequesters a similar amount of carbon through its sinking faecal pellets as marshes, mangroves and seagrass. Due to their massive population biomass, fast-sinking faecal pellets and the modest depths that pellets need to reach to achieve sequestration (mean is 381 m), Antarctic krill faecal pellets sequester 20 MtC per productive season (spring to early Autumn). This is equates USD\$ 4 – 46 billion depending on the price of carbon, with krill pellet carbon stored for at least 100 years and with some reaching as far as the North Pacific. Antarctic krill are being impacted by rapid polar climate change and an expanding fishery, thus krill populations and their habitat warrant protection to preserve this valuable carbon sink.

Marine life has an important role in locking carbon away from the atmosphere in ocean systems, known as 'blue carbon'. Coastal vegetation such as seagrass, mangroves and salt marshes dominate blue carbon research and policy[1,2]. Another important form of carbon sequestration which is ubiquitous in the global oceans is through the open-ocean biological carbon pump which stores carbon in the deep-sea[3], and at present is not regularly framed in the blue carbon context. The biological carbon pump works by phytoplankton fixing dissolved $CO_2$ into organic carbon during photosynthesis; when the cells die, they may sink to the deep ocean, along with the faecal pellets of phytoplankton grazers (zooplankton) and higher organisms such as fish[4]. If the sinking carbon is not remineralised, it may be sequestered for decades to centuries[5], such that without this biological pump there would be 50 % more $CO_2$ in the atmosphere[6,7].

A key player in the biological carbon pump is the Antarctic krill, *Euphausia superba*, whose faecal pellets, moults and carcasses can completely dominate Southern Ocean carbon fluxes to the ocean's interior[8–14] in the Austral growth season (October through to April, which is our study period). Krill face human-induced threats; through climatic warming reducing sea-ice[15] with record Antarctic sea-ice loss this year[16], which is an important habitat for larval krill[17]. An additional

issue is the ongoing expansion of the krill fishery in the absence of a management system that accounts for temporal and spatial variability in krill ecology[18]. Antarctic ecosystem services are globally important but these services are poorly understood in comparison to those elsewhere in the world[19]. Quantification of metrics that allow global comparisons is an important step towards closing this information gap and ensuring that decision-makers understand the importance of these services. Here we aim to address this by estimating carbon sequestration for the entire circumpolar Antarctic krill population. This advance will allow comparison with coastal vegetation blue carbon systems, showcasing the relevance of open-ocean animals in the same context. The approach used here can also be applied to other pelagic organisms (such as forage fish, jellyfish or gelatinous zooplankton), advancing our understanding of pelagic life in sequestering carbon.

The publication of a circumpolar Antarctic krill density database spanning 90 years (KRILLBASE[20]) facilitated the first estimates of krill faecal pellet carbon production (FPCprod) and subsequent export from the surface ocean on a large spatial scale. Belcher et al. [10] estimated that in just the marginal ice zone during the productive spring to early autumn months, krill FPCprod and export from the surface

[1]Imperial College London, Ascot, Berkshire, UK. [2]University of Exeter, Exeter, Devon, UK. [3]British Antarctic Survey, High Cross, Cambridge, UK. [4]Plymouth Marine Laboratory, Prospect Place, The Hoe, Plymouth, UK. [5]UK Centre for Ecology and Hydrology, Bush Estate, Midlothian, UK. [6]VKR Centre for Ocean Life, Technical University of Denmark, 2800 Kongens, Lyngby, Denmark. ✉e-mail: e.cavan@imperial.ac.uk

ocean (< 100 m) is ~40 MtC yr$^{-1}$ with export greatest where krill numbers are highest (often in the Atlantic Southern Ocean). However, carbon export is not equivalent to carbon sequestration. In order to become sequestered, exported carbon must avoid remineralisation and sink into a water mass that will not be upwelled or ventilated elsewhere in the ocean for at least 100 years[21].

In this study, we use krill density[20] multiplied by our assessed faecal pellet egestion rate for Antarctic krill (see Supplementary Note S2 and Supplementary Tables S1–3) to estimate a faecal pellet carbon production flux (FPCprod) in the upper ocean (~20 m). We use this as our 'export' flux and attenuate this to the depth the carbon would need to reach to remain in the ocean for at least 100 years. We take sequestration depth data from an ocean circulation model[22]. This results in our krill faecal pellet carbon sequestration flux (FPCflux), defined as pellet carbon flux at an ocean depth that is sequestered for at least 100 years. To calculate the FPCflux, we attenuate the FPCprod flux following a power-law function known as 'Martin's b', where the 'b' attenuation coefficient is −0.3 for krill, a lower attenuation than the global average due to the fast-sinking nature of krill pellets (see "Methods"). We then use the Social Cost of $CO_2$ to convert our krill FPCflux estimates into a single comparable metric: USD ($), and compare it to other, better-known sequestration sources such as mangroves, seagrasses, and salt marshes. Although other life-history traits of Antarctic krill can be important in carbon fluxes, such as their moults and active $CO_2$ transport from migrations, the data and knowledge available to investigate spatial and temporal patterns is greatest for krill faecal pellets, and hence this is our focus. We find Antarctic krill faecal pellets can sequester 20 MtC over the productive Austral spring and summer seasons. This single source of sequestration, from just a single species, is broadly similar to global totals for much more widely documented blue carbon habitats, such as mangroves, salt marshes and seagrasses.

## Results

### Circumpolar krill faecal pellet carbon sequestration flux

To estimate the circumpolar sequestration of krill faecal pellets, we combined four key elements and here map the data for the month of January, namely a climatological mean krill distribution (Fig. 1a); literature estimates of their faecal egestion to give mixed layer Krill FPCprod fluxes; the rate of attenuation of this flux with depth (here *Martin's b = −0.3*); and the depth needed to achieve sequestration (Fig. 1b). The latter 'carbon sequestration depth' is defined as the water column depth where the time-to-surface of the water is at least 100 years, calculated using an ocean circulation inverse model (OCIM)[22]. The OCIM is a global coarse-resolution (2°) model with 24 vertical levels, for which a steady-state physical circulation has been optimised for consistency with observed passive tracer distributions, sources and sinks, using an adjoint method[23]. The carbon sequestration depth ranges from 135 to 760 m and is deepest in the Indian Ocean. However, the spatial pattern of estimated carbon sequestration (Fig. 1c) was predominantly driven by the distribution of krill density, with the highest values in the SW Atlantic sector. The mean carbon sequestration depth for the whole data set was 381 m (Table 1), suggesting that particles only need to sink into the mid-mesopelagic zone of the Southern Ocean to remain sequestered for >100 years. Therefore, some upper ocean observations of sinking particulate organic carbon flux from krill could have observed fluxes that were contributing to carbon sequestration. In all but two of the spatial cells (0.4 % of the total), the length of time carbon is stored increases monotonically with depth. In some of the krill hotspots, such as the Scotia Sea (Atlantic Sector), the carbon sequestration depth is fairly shallow, and a faecal pellet may only need to sink to ~200 m before becoming part of a water mass that would remain away from the surface ocean for 100 years. This highlights how important the choice of depth horizon is in flux observations and

models, suggesting that arbitrary depth thresholds might bias estimates of carbon sequestration in the ocean[24,25].

Nominal krill FPCprod flux at 20 m depth ranged from 0 – 276 mgC m$^{-2}$ d$^{-1}$ with the highest mean flux in the Atlantic sector of the Southern Ocean (and Table 1), as this is where Antarctic krill are most densely populated (Fig. 1a). These krill pellet export fluxes are similar in magnitude to observed zooplankton and krill pellet export rates in the Southern Ocean[11]. We use a lower egestion rate of 0.46 mgC krill$^{-1}$ d$^{-1}$ compared to earlier studies such as Belcher et al.[10], who used a rate of 3.2 mgC d$^{-1}$ and hence our range is an order of magnitude smaller (~300 compared to ~3000 mgC m$^{-2}$ d$^{-1}$). Our revised egestion rate estimate is derived from a number of independent methods (Supplementary Tables S1–S3) and is intended to reflect the range of feeding conditions experienced between spring to autumn and at the circumpolar scale, whereas the higher values in the literature represent short observation periods during a time of high food availability[26].

The krill pellet export fluxes at 20 m were attenuated to simulate the degradation of pellets with depth due to consumption and bacterial remineralisation using a power-law coefficient[27] based on observations from krill pellet fluxes presented in Belcher et al.[13] and including more recent krill pellet fluxes from Pauli et al.[28] (see Supplementary Table S4). This results in a krill pellet attenuation rate (*Martin's b*) of −0.30, which shunts more carbon to depth than the 'global' value of b of −0.86 commonly applied to all sinking POC types. The lower attenuation of krill pellets results mainly from the rapid sinking of these pellets[9,29]. In addition, our attenuation rate is an average derived from observations which compare fluxes at mixed layer and mesopelagic depths and are likely affected by additional egestion between these two depths[11]. In the absence of detailed information on the depth distribution of egestion, this value appropriately compensates for the simplifying assumption in our calculation that faecal pellet carbon is only attenuated between 20 m and the carbon sequestration depth, rather than also being produced.

### Sequestered krill faecal pellet carbon

Krill FPCflux at the sequestration depth ranged from 0 – 120 mgC m$^{-2}$ d$^{-1}$ (Fig. 1c), remaining highest on average in the Atlantic sector (Fig. 1c and Table 1). The total carbon that could be sequestered each Austral spring/summer was estimated by summing the pellet carbon fluxes (FPCflux, Fig. 1c) at the carbon sequestration depth over the area (m$^{-2}$, Fig.1) and the number of days in the months (October to April) sampled. Total nominal krill FPCprod from within the mixed layer was 44 MtC yr$^{-1}$, with 20 MtC yr$^{-1}$ (45 %) sequestered (Fig. 2). This sequestered carbon equates to a range in monetary value of USD$4 – 46 billion (Table 1, Fig. 2), depending on the value of the Social Cost of $CO_2$ per tonne of $CO_2$ used.

Following the trend with the krill densities and pellet fluxes, most of the carbon is injected into the interior ocean in the Atlantic sector of the Southern Ocean, where the carbon sequestration depth is fairly shallow. Using spatially-resolved net primary productivity (NPP) values for the Southern Ocean[30], we find that on average ~2.5 % of NPP is routed to krill pellet carbon sequestration (i.e., FPCflux/NPP) across the Southern Ocean, with values being up to 74 % in swarm locations (see Supplementary Table S5). The 'transient time',i.e., the time from photosynthesis by phytoplankton capturing dissolved inorganic carbon to krill pellet carbon being sequestered is likely less than a week, considering gut passage times of less than a day and pellet sinking rates of ~300 m d$^{-1}$[29]. This shows how, in particular, krill swarms are extremely efficient vectors of carbon sequestration, quickly grazing down phytoplankton and the subsequent mass production of fast-sinking faecal pellets[10]. The proportion of krill FPCprod that is sequestered (i.e., the transfer efficiency) is much higher (45 %) than the commonly reported 1–10 % in the biological pump because of the low attenuation rate of krill pellets and because pellets only need to sink a

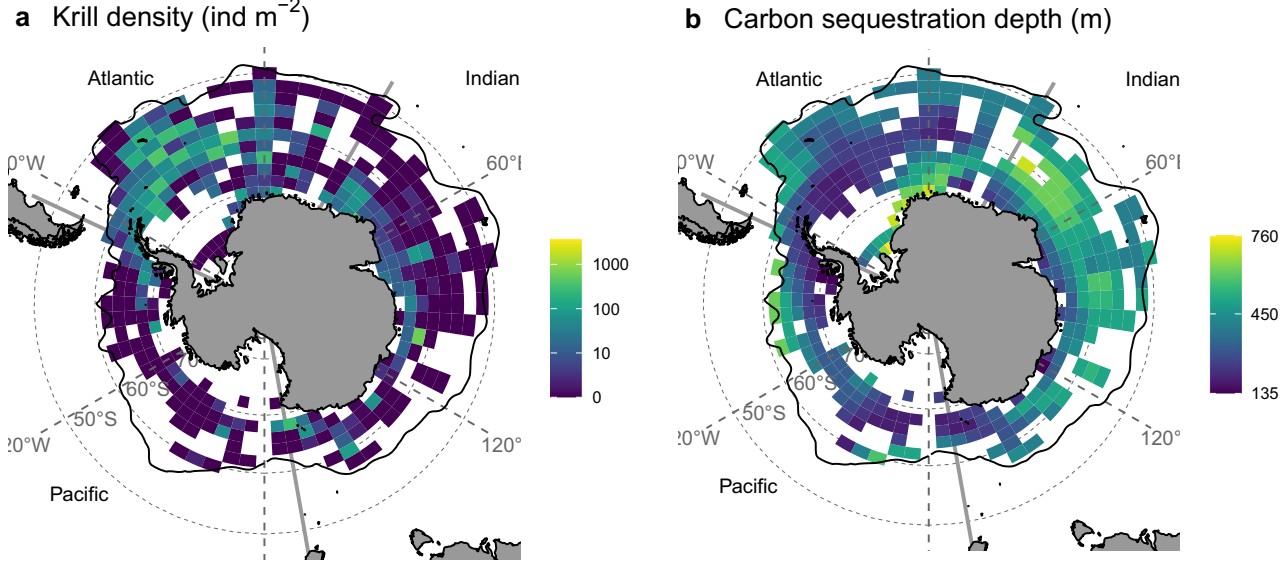

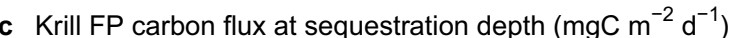

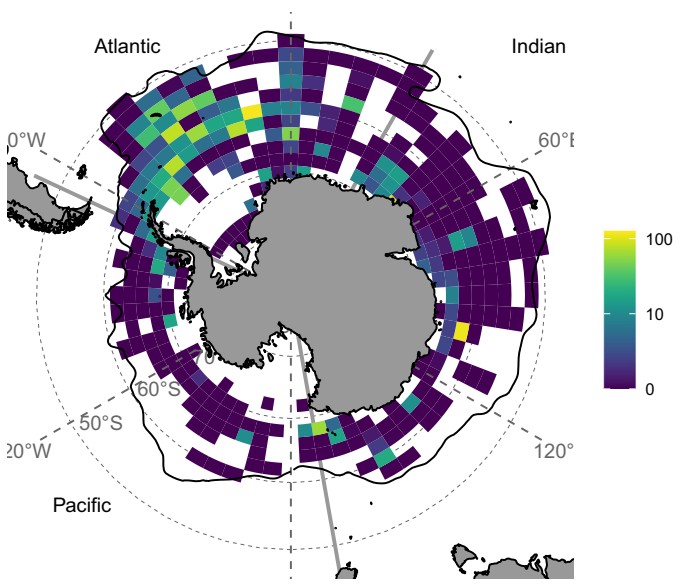

**Fig. 1 | Circumpolar krill faecal pellet carbon sequestration flux.** The spatially varying parameters (**a**) krill density (ind m⁻², January mean for data spanning 1926–2016) and (**b**) carbon sequestration depth, used along with egestion and attenuation rate to give (**c**) nominal krill faecal pellet carbon sequestration flux (FPCflux), attenuated to the carbon sequestration depth. The resolution is 2° latitude by 6° longitude. Note: the carbon sequestration depths are assumed constant in time (**b**), but data in (**a**, **c**) are for the month of January. See Supplementary Fig. S1 for the circumpolar densities per month for the entire time series. The solid black line represents the location of the Antarctic Polar Front.

few hundred meters to achieve sequestration, compared to the 1000 m or 2000 m depths often used to define 'transfer efficiency'[22,31]. This contributes to the growing body of work moving biological carbon pump research on from calculating carbon fluxes at arbitrary discrete depths, to factoring in carbon sequestration timescales[25].

To determine the additional contribution of sequestered krill pellet carbon flux relative to plankton (phytoplankton and copepods) typically represented in biogeochemical and biological pump models, we use a published food-web model output[32] (updated from ref. 22) to estimate the total carbon sequestered by plankton in our study region. The model by Nowicki et al. [32] estimates that circumpolar phytoplankton and copepods sequester 160 MtC compared to the 20 MtC we estimate for Antarctic krill, meaning that krill faecal pellets account for 12 % of the total 'plankton' carbon sequestration in the Southern Ocean.

This 12 % likely represents carbon sequestration incorrectly allocated to phytoplankton detritus or copepod faecal pellets in models that do not explicitly represent such large swarming organisms as krill. Therefore, our type of model is important for conservation purposes as it can highlight the locations of high pulses of carbon to the ocean interior (see Supplementary Fig. S2a) from organisms of interest, here krill. When we include our best estimates of the additional carbon sequestration potential from krill moults (20 MtC) and active transport (26 MtC) (see Methods for details), then krill could sequester a total of 66 Mt Cy⁻¹. The current generation of Earth System Models (ESMs) underestimate or entirely omit the contribution of these larger organisms to the ocean carbon cycle. Incorporating micronekton, such as krill, and behaviour, such as diel vertical migration, should be priorities for biological parameter development in ESMs.

**Table 1 | Mean and ranges of krill faecal pellet production (FPCprod) flux in the mixed layer and mean krill FPCflux at the carbon sequestration depth for all months, October to April, across the entire spatial dataset (All) and by Southern Ocean region (Fig. 1)**

| | Mean | | | Total | |
|---|---|---|---|---|---|
| Region | FPCprod (mgC m⁻² d⁻¹) | FPCflux (mgC m⁻² d⁻¹) | C sequest. depth (m) | C sequestered (MtC) | (billion USD $) |
| Atlantic | 14.9 (0–253) | 6.8 (0–120) | 373 (188–758) | 13.7 | 2.6–32.2 |
| Indian | 7.1 (0–276) | 2.9 (0–112) | 449 (137–712) | 4.1 | 0.8–9.5 |
| Pacific | 3.6 (0–129) | 1.8 (0–64) | 313 (137–620) | 1.9 | 0.3–4.4 |
| All | 9.3 (0–276) | 4.2 (0–120) | 381 (137–758) | 19.7 | 3.7–46.1 |

Total carbon sequestered in MtC and USD is also given per region and summed over all regions. The range of carbon sequestered reported in USD ($) is given by applying the value of carbon as $51 per tonne and $640 per tonne (see Methods). The sequestered carbon flux (FPCflux) in each cell is a function of krill density, with density varying spatially and temporally (range = 0–600 ind. m⁻²) and driving the ranges in carbon fluxes.

## Validating our approach

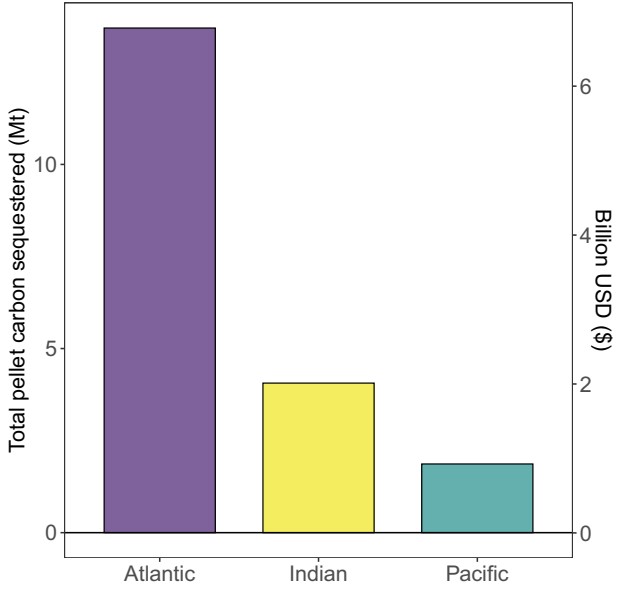

**Fig. 2 | Total carbon sequestered by krill faecal pellets.** Total carbon sequestered in Mt C (million tonnes) and billion USD per sector of the Southern Ocean. In this graphical representation, an economic value ($135 per tonne of $CO_2$ sequestered) at the lower end of the considered range ($51–640) is used. The total across all three Southern Ocean regions is 20 Mt C and USD $4–46 billion using the full range of the Social Cost of $CO_2$. See Table 1 for the ranges in fluxes that lead to the total carbon sequestered presented here and Methods for details on uncertainty and error.

Our total pellet carbon sequestration estimates (Fig. 2) are based on detailed information on the distribution and density (number of krill individuals per unit area) of krill[15], information on carbon sequestration depth and a revised set of parameters to represent egestion rate and the attenuation rate (*Martin's b*) of sinking faecal pellets. Our values for each parameter are within the range of values reported in the literature (see "Methods" and Supplementary Tables S1–S4). Our revised egestion rate is more conservative (i.e., results in lower *per capita* krill FPCprod fluxes than previously used[10]), whereas our attenuation rate is based on krill pellet flux observations[10], and because krill pellets sinking rapidly, the attenuation is more gradual than the value typically used to model global total POC flux (i.e., it preserves more of the carbon to sequestration depth). Our circumpolar krill density ($5.7 \times 10^{14}$ krill individuals in 16 million km² of

Southern Ocean) is similar to a previously reported circumpolar krill density of $5.4 \times 10^{14}$ krill[33]. We compare our model with krill pellet carbon observations in our study area to assess whether our methodology is appropriate and representative of krill pellet fluxes.

Our model compares well with direct observations of krill faecal pellet carbon and total particulate organic carbon (POC) fluxes measured in the Southern Ocean. First we compare to South Georgia (53°S 38°W) sediment trap data at 300 m depth[13]. Extracting the krill faecal pellet flux in January from the annual sediment trap time series in Manno et al. [13] results in krill pellet flux at 300 m of 46 mgC m⁻² d⁻¹. We attenuated krill FPCprod fluxes from our model to 300 m at the location of the South Georgia sediment trap, which gave a similar krill FP flux of 40 mgC m⁻² d⁻¹. We also compare our model to observations of total POC fluxes further south at the Palmer Station (64°S 66°W) on the Western Antarctic Peninsula[14]. Here the mean total POC flux (pellets plus other sinking particles) in January from a 20-year time-series sediment trap at 170 m depth was 16 mgC m⁻² d⁻¹. Our modelled krill pellet flux at this location and depth is 6 mgC m⁻² d⁻¹, suggesting that 38 % of the total POC flux is made up of adult krill pellets near the sea ice (Supplementary Fig. S2b). Krill pellets tend to dominate the Palmer Station sediment trap in Austral summer, contributing up to 86 % of the total observed POC flux[12]. Our value of 38 % is lower than this because our model only accounts for adult krill > 40 mm, and close to the sea-ice and continent, there is a much larger population of juvenile krill, compared to, for instance, South Georgia[15]. These results give confidence that our large-scale estimates of krill carbon fluxes are reasonable and within the expected ranges based on observations.

### Uncertainty in our estimates

We ran a sensitivity analysis, varying each parameter in turn by ± 10% (Fig. 3a, Supplementary Table S6) to identify the relative influence of each on our estimate of total carbon sequestered (as in Fig. 2). These parameters were krill density, egestion rate, attenuation rate and sequestration depth. Krill density and egestion rate have the joint largest influence. These two parameters together provide the krill FPCprod flux which sets the amount of carbon that can be injected into the deep ocean. As krill density is the main forcing data for our model and it has a large range over orders of magnitude (0–600 ind. m⁻²), krill distribution sets the spatial spread of the carbon fluxes (Fig. 1). The carbon sequestration depth is the other forcing data used in our model but had the least influence in our sensitivity analysis. Whilst it does vary regionally (Fig. 1b) the changes in sequestration depth are more gradual and less pronounced, hence in terms of forcing data krill density has the largest impact on carbon sequestration.

We conducted an additional analysis, varying each parameter based on the range of plausible values suggested by our analysis (for krill density, egestion rate and carbon sequestration depth) or observations reported in the literature (for *Martin's b*). This allows us to identify where the scientific understanding is least certain and/or where natural variability is highest, and more research is needed to further constrain these parameter estimates. Here, the greatest influence on total carbon sequestered is the huge range of *Martin's b* observations (−2.46 to 1.81) (Fig. 3b and Supplementary Table S4). This range reflects both the spatial and temporal variability in attenuation of pellets, limitations with measurements which can arise from the time delay between pellets leaving the surface and reaching the mesopelagic[34], and egestion beneath the assumed export depth[8,35]. We consider *b* values of − 0.61 and + 0.13, which are the 25th and 75th percentile of values from the literature (see Supplementary Table S4) and result in total carbon sequestered of 9 and 63 MtC, 44 % or 320 % of our best estimate (20 MtC) respectively (Fig. 3b and Supplementary Table S6). The total carbon sequestered with *b* set to the 'global' accepted value of − 0.86 is 4 Mt C. The next largest effect,

### a  Parameter sensitivity analysis

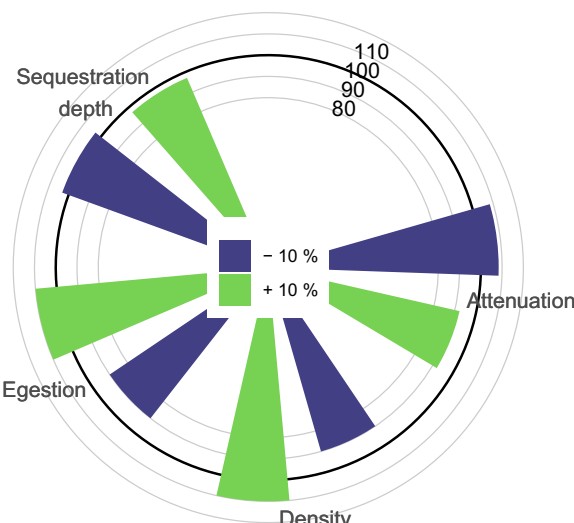

### b  Uncertainty in estimates

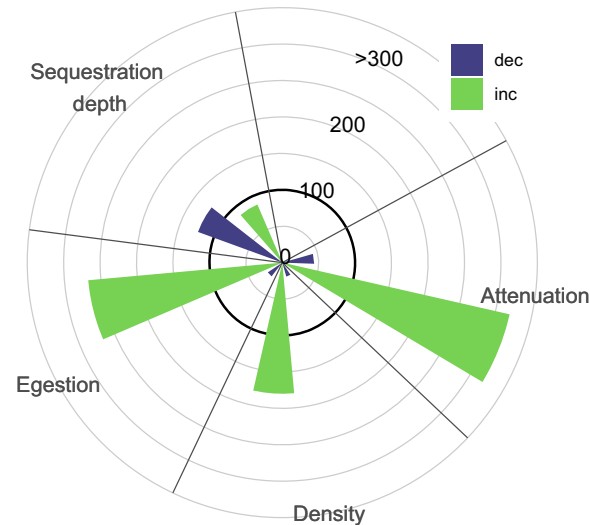

**Fig. 3 | Sensitivity analysis. a** parameter sensitivity analysis changing each parameter by 10 %, and (**b**) uncertainty analysis where parameter values are changed based on observations reported in the literature. The circular gridlines give the percentage change in total carbon sequestered, with 100 % equivalent of 20 Mt C. Parameters which set the rate of export of pellet carbon out of the surface ocean, krill density and egestion rate, have the largest effect on overall carbon sequestered

(Fig. 3a). However, the largest natural variability and/or variation in observations to date is associated with the attenuation of pellets (**b**) with depth (Fig. 3b), even though the 25th and 75th percentiles are represented for attenuation, whereas more extreme values (5th and 95th percentile) are used for egestion and sequestration depth (see "Methods").

increased egestion rates, results in a range of total carbon sequestration values from 24 to 267 % of our best estimate (Fig. 3). To improve modelling studies such as ours, we need to understand what factors control the pellet attenuation rate, with candidates including krill diet, composition of mesopelagic consumers and water temperatures. This will enable spatial and temporally varying attenuation rates to be applied.

**Tracking krill carbon through the oceans**

Carbon transferred to the deep by Antarctic krill does not necessarily stay in the Southern Ocean. As bacteria break down the carbon-rich pellets, they produce dissolved inorganic carbon, which can be transported around the global oceans. To track the fate of krill pellet carbon after it has been injected into the ocean's interior, we ran an additional model analysis using an Ocean Circulation Inverse Model (OCIM; see "methods")[22]. This allows us to determine the steady-state interior distribution of dissolved inorganic carbon originating from krill faecal pellets, i.e., the total amount of carbon stored from krill pellets in the ocean at any one time. Most of the krill carbon remains in the Atlantic sector (Fig. 4), where krill pellet carbon fluxes are highest (Figs. 1, 2); meanwhile, some are transported to the Pacific and Indian Oceans, including in the Northern Hemisphere with very little reaching the North Atlantic Ocean. Once krill pellet carbon leaves the surface ocean in the form of krill faecal pellet fluxes, it remains there with an average residence time of 219 years before it comes into contact with the surface again. Based on our krill pellet export fluxes and attenuation rate ($b = -0.30$) the total amount of carbon derived from Antarctic krill pellets that the oceans can hold is estimated as 8.7 Gt C (Fig. 4). This attenuation rate results in ~20 % of the export carbon reaching the benthos as particulate pellets, where additional carbon sequestration mechanisms such as burial and incorporation into the bodies of long-living invertebrates can occur[36]. The total krill-carbon stored in the global water column reduces to 1.7 Gt C for only 58 years if the non-krill pellet attenuation rate of $b = -0.86$ is applied instead (Supplementary Figs. S3, S4). This emphasises how important

the remineralisation rate, or the attenuation rate, is in estimating long-term carbon storage by sinking particles. Given the evidence that for krill pellets, the carbon is shunted deeper into the ocean than other types of particles (Supplementary Table S4), the $b$ parameter should vary with particle type in model simulations.

## Discussion

### Krill in the context of blue carbon

Defining sinking krill pellet fluxes in the context of the physical circulation allows us to quantify the carbon sequestered (defined here as locked away for >100 years), and thus discuss them in the context of other blue carbon stores. Our estimated sequestration store of Southern Ocean krill faecal pellets (20 MtC) over an Austral productive season is of the same magnitude as other global blue carbon stores, with saltmarsh, mangroves and seagrass estimated to sequester 13, 24 and 44 MtC yr$^{-1}$[1], respectively (Fig. 5). The krill-pellet carbon injected into the oceans is three orders of magnitude lower in concentration (1.3 tC km$^{-2}$ yr$^{-1}$) than for coastal vegetative blue carbon stores that predominantly store carbon in highly concentrated soils and sediments (e.g., seagrass = 156 tC km2 yr$^{-1}$)[1] (Fig. 5). It is the vast Southern Ocean habitat that Antarctic krill occupy (16 million km$^2$ – our study – to 19 million km$^2$[33]) compared to coastal vegetation (e.g., seagrass = 0.32 million km$^2$ globally) that makes the total carbon sequestered by krill equal to that of the coastal blue carbon stores. Given our conservative approach to faecal pellet export estimation, the significant quantities of carbon in living krill[37] and the additional flux that will result from krill moults (~20 MtC), winter feeding and respiration during migrations (~ 26 MtC)[4,12,13] (Fig. 5) it is likely that Antarctic krill is amongst the world's most important carbon-storing organisms. Given the comparability of these numbers associated with Antarctic krill to other carbon sequestration mechanisms, we suggest our approach can be used to quantify the carbon sequestration value of other pelagic marine life, such as copepods, gelatinous zooplankton and potentially low-mid trophic level nekton such as forage fish.

## Dissolved inorganic carbon (gC m$^{-2}$), b = −0.30

**Fig. 4 | Global pelagic carbon footprint from Antarctic krill faecal pellets.** Equilibrium dissolved inorganic carbon (DIC) concentrations (gC m$^{-2}$) from the remineralisation (attenuation) of Antarctic krill faecal pellets with depth, assuming a remineralisation coefficient of $b = -0.30$. The highest water-column concentrations exist in the Atlantic Southern Ocean, where krill pellet fluxes are highest (Figs. 1, 2). This map does not include the remaining pellets that escape remineralisation and fuel Antarctic communities on the seafloor.

### Conservation perspectives

The krill-dominated habitats of the Southern Ocean are akin to the better-documented salt marsh, mangroves and seagrass habitats: not only do they support unique, valuable and iconic species, but they are also key conduits for carbon storage. Both of these facets warrant the utmost importance of conserving into the future. The value for krill is in the order of billions of dollars in terms of carbon storage that benefits society as an ecosystem service. Krill are under threat from warming resulting in loss of habitat, potential changes to food availability and competition, and from the krill fishery[15,19]. Whilst there are some fishery restrictions and protected areas in place in the Southern Ocean, more needs to be done to ensure this carbon sink can remain active into the future. It is not clear how a different upper ocean community (i.e., with less krill) may look, and the implications this has for carbon sequestration. Though other low-trophic level pelagic organisms are increasing in numbers in the Southern Ocean (e.g., salps)[38], their efficiency in transferring carbon deep enough for sequestration is not always as high as krill. For instance, salp pellets do not always penetrate into the deeper ocean in such high numbers due to their fragility[28,39], compared to dense krill faecal pellets. These subtleties illustrate the complexity of the biological carbon pump.

Antarctic krill are clearly important vectors of carbon sequestration. The large uncertainty in our understanding of future pelagic communities and carbon export under climate change means we cannot predict with certainty how future changes in the krill population will affect the magnitude of carbon sequestration in the Southern Ocean[40]. Our study highlights the importance of Antarctic krill, and likely other components of the biological pump, in sequestering carbon which is on par with coastal blue carbon stores. This study and others[14,41,42] are increasingly showing the importance of krill and other zooplankton in structuring Southern Ocean food webs and activating a strong biological pump. Understanding the relative roles of the key phyto-, zooplankton and micronekton functional groups in carbon sequestration is therefore an urgent priority. Valuing pelagic vulnerable marine ecosystems in terms of carbon sequestration, like that of Antarctic krill, emphasises how crucial it is to meet climate goals and work towards including carbon policies in resource management.

## Methods

### Krill density

Krill density data were obtained from KRILLBASE[20], which compiles postlarval krill numerical densities (number per unit sea surface area) based on net sampling around Antarctica spanning 1926–2016. These data were filtered following previously published procedures[15], where only data collected from 'hauls' or 'stratified pooled hauls' where the top sampling depth was shallower than 20 m, and the bottom sampling depth deeper than 50 m were used. The data set contains a column of 'standardised krill under 1m2', which represents krill density normalised to the first of January, using efficient sampling gear during the night-time (when krill are likely to be in the surface layers). This standardisation uses empirical algorithms to extrapolate what the krill density would be on the 1st of January, depending on when the sample was collected (Table 4 of Atkinson et al. [20], and Supplementary Equations S1–S3). This allows for almost full circumpolar coverage of krill density for the month of January. The same model can, therefore, be used to calculate what the krill density would be on the first date of each month (Supplementary Fig. S5), resulting in krill density estimates for the full Austral spring-summer season, from October to April (Supplementary Fig. S1). See Supplementary Note S1 and Supplementary Equations S1–S3 for more information.

The data were then projected onto a 2° × 6° lat/lon grid, by averaging (mean) krill densities ($\bar{N}$) in each month ($t$) over this spatial scale. This is more conservative than a 3 × 9° grid (Supplementary Fig. S6) and in line with a resolution also used by Atkinson et al. [33]. It balances the risk of inappropriate extrapolation (when data are only available for a small fraction of a grided square), with the risk of underrepresenting the extent of krill habitat (as would occur with the 2° × 2° resolution of OCIM) (Supplementary Fig. S7). Where krill densities were > 600 ind. m$^{-2}$ ($n = 7$, 0.3 % of data, ranged from 673 – 1681 ind. m$^{-2}$) these were capped at 600 ind. m$^{-2}$. This excludes bias from a

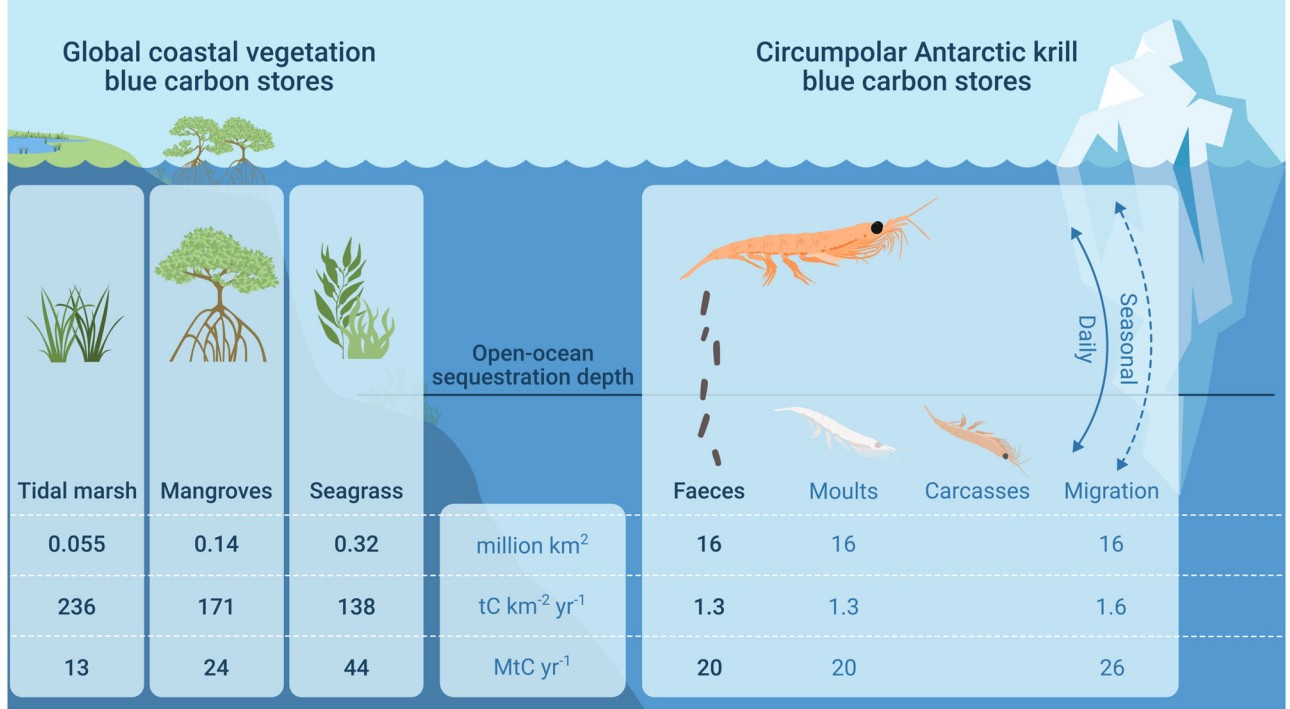

**Fig. 5 | Comparison of krill pellet blue carbon with coastal vegetation blue carbon stores.** The krill pellet carbon sequestered in MtC per October-April growing season is given and converted to per km². Krill sequester less per surface area of the ocean than coastal vegetation blue carbon, but the vast area krill inhabit puts the pellet carbon sequestered in the same order of magnitude as the coastal vegetation stores, according to data in Bertram et al. We also include best estimates of the contribution of sinking moults and daily migration (active transfer of $CO_2$). If these are correct, krill could sequester a similar amount of carbon as seagrasses do globally. There is insufficient data to yet do this analysis for krill carcasses, but we include them graphically as they are an important carbon sink. Note, that other krill habitat size estimates find krill habitat to be 19 km² using a different grid resolution.

few extremely high net catches sampling in a dense swarm but still allows for high krill densities to occur, representing regions where swarms are commonly found. The 7 data capped (one for each month) were in fact, from just one net haul, and removing data from this observation reduced the estimated total carbon sequestered from 21.5 MtC yr⁻¹ to our value reported of 20 MtC yr⁻¹.

**Faecal pellet flux**

For each month ($t$), krill faecal pellet carbon production flux (FPCprod) at 20 m for each $2 \times 6°$ cell ($k$) was calculated by multiplying the monthly mean krill density ($\bar{N}$, # m⁻²) by a krill faecal pellet egestion rate ($E$) (Eq. 1).

$$FPCprod_{k,t} \left( mgC\,m^{-2}\,d^{-1} \right) = \bar{N}_{k,t} \cdot E \qquad (1)$$

High daily rates determined from laboratory studies used in some previous studies (e.g., 4.03 mg C ind⁻¹ d⁻¹ (ref. 26) and 3.2 mg C ind⁻¹ d⁻¹ (ref.10)) are unlikely to be sustained throughout the summer season or at the circumpolar scale. Assuming higher egestion rates, as in these previous studies, would have commensurate effects on the estimated magnitude and value of krill-mediated carbon sequestration, as shown in our sensitivity analysis. We, therefore, used three different methods to estimate egestion rates, and we assessed their validity by converting them to estimates of annual food consumption (in carbon units) by the circumpolar krill stock and comparing this to an indicative estimate of circumpolar primary production (1949 Mt C y⁻¹)[43]. The resulting median egestion rate for an adult krill of 40 mm length was 0.46 mg C ind⁻¹ d⁻¹, with a 5-95th percentile range of 0.11 to 1.23 mg C ind⁻¹ d⁻¹. The results of this analysis and more detailed methods are available in Supplementary Note S2 and Supplementary Tables S1–S3.

The mesopelagic krill FP carbon flux (FPCflux) was calculated using Eq. 2, which attenuates the krill FPCprod between 20 m ($z0$) and any mesopelagic depth ($z$) using *Martin's b* value of −0.30, based on data collated by Belcher et al.[10] with the addition of Pauli et al.[28]. As the attenuation rate is a crucial factor in determining pelagic sequestration, we also repeated the calculation using the globally applied estimate of −0.86, increasing the remineralisation of the FPs as they sink to give a more conservative estimate. As *Martin's b* describes the rate of transfer of POC flux to depth, it implicitly represents the sinking rate, remineralisation rate and zooplankton fragmentation rates of sinking faecal pellets.

$$FPflux_{k,t,z} \left( mgC\,m^{-2}\,d^{-1} \right) = FPCprod_{k,t} \cdot \left( \frac{z}{z0} \right)^b \qquad (2)$$

**Carbon sequestration depth**

The main purpose of this study was to calculate the amount of FP flux that would remain sequestered in the oceans for at least 100 years, and thus ocean circulation must be considered. For this, we used the Ocean Circulation Inverse Model (OCIM) model output product from DeVries & Weber[22], which outputs the time-to-surface of a parcel of water throughout the water column, known as the First Passage Time (FPT). This allows us to find the depth for each cell at which the FPT would be at least 100 years, and to conclude that if the FP sank below this depth, then the carbon, either as particulate organic or dissolved inorganic carbon (POC or DIC) would remain away from the surface of the ocean for 100 years or more and is therefore sequestered.

We calculated the depth where FPT = 100 years or more, and this became our sequestration depth or FPT₁₀₀. In the OCIM model, the sequestration times (years) are based in part on a winter mixed layer depth, as anything sinking shallower than that would be ventilated within a year. FPT in the ocean does not increase monotonically with depth due to overlapping water masses. However, for the OCIM

outputs used in this study, FPT was found to increase monotonically with depth in all but two horizontal grid cells. This result is somewhat unexpected, given, for example, the structure of water masses in the Southern Ocean, where upwelling Circumpolar Deep Water sits underneath downwelling mode and intermediate waters, potentially highlighting weaknesses in the coarse resolution OCIM's representation of the circulation. If the analysis were carried out using a finer resolution physical model, the final sequestration could decrease in some places or increase in others; unfortunately, no finer resolution version of the OCIM currently exists, and analysis of the sequestration of krill carbon using a forward model is outside the scope of this study.

To determine the krill FP carbon sequestration (FPCflux), we applied *Martin's b* curve again, but this time only for the $FPT_{100}$ depth, which varied in space ($k$) but not in time:

$$FPCflux_{k,t}\left(mgC\,m^{-2}\,d^{-1}\right) = FPCprod_{k,t} \cdot \left(\frac{FPT100_k}{z0}\right)^b \quad (3)$$

The final total circumpolar carbon sequestered by krill FPs (Table 1 and Fig. 2) was found by multiplying FPCflux by the area of each cell ($A_k$, m$^{-2}$) and the number of days in the month ($m_k$), and summed across the whole sampling area ($k$) and months analysed ($t$) (October to April):

$$FPseq_{tot}\,(GtC) = \sum_{k,t}\left(FPCflux_{k,t} \cdot A_k \cdot m_t \cdot 1e^{-18}\right) \quad (4)$$

Error bars are not presented in Fig. 2 as the final data here are sums, not means, and the main forcing data of krill densities are not normally distributed. Instead in Table 1 we report the ranges of fluxes present in our data, which lead to the total summed carbon stored. The statistical error can only be calculated based on krill density means and the associated error, such as standard deviation ($20 \pm 62$ ind. m$^{-2}$). The variation in the mean is large as the range in krill density is large (0–600 ind. m$^{-2}$), with a median krill density of 1 ind. m$^{-2}$. Calculating uncertainty in our carbon sequestration results based on krill densities leads to negative carbon sequestration values. We, therefore, report the ranges of fluxes in Table 1 and run sensitivity and uncertainty analyses (see below and Fig. 3).

## Sensitivity analysis
The relative influence of each parameter used to estimate the final carbon sequestered (GtC) was investigated by sequentially increasing or decreasing each parameter by 10 % and investigating the impact on the final sequestered carbon at the sequestration ($FPT_{100}$) depth for krill faecal pellets. A second analysis varied each parameter sequentially by either the 5th and 95th percentile values of those in our data or analysis (egestion rate and sequestration depth). For the attenuation rate (*Martin's b*), we compiled observations of krill pellet fluxes from the literature but restricted the percentiles to the 25th and 75th in our uncertainty analysis as the effect of changing attenuation was large (Fig. 3b). Finally, for krill densities as the data were so skewed and over a large range, we instead used as the upper limit the arithmetic mean of all KRILLBASE records reported by Atkinson[33] of 36 ind. m$^{-2}$ and centred our mean (20 ind. m$^{-2}$) so the lower limit was 4 (i.e., ±16 ind. m$^{-2}$).

## Economic value
The total sequestered krill faecal pellet carbon (Eq. 4) was then converted from mass in Gt to its worth to society in monetary terms by multiplying by the Social Cost of $CO_2$ ($SCCO_2$, Eq. 5). At the time of writing the US was still using their interim $SCCO_2$ of only USD\$51 per t$CO_2$, but a 2021 study found the global average $SCCO_2$ was USD\$640 per t$CO_2$[1]. However, a 2018 IPCC report predicts values could range from USD\$135 – 5500 per t$CO_2$ by 2030 to reach only 1.5 °C of

warming[44]. Here we use a range of $SSCO_2$ from USD\$51 – 640 per t$CO_2$. To convert from the monetary value of $CO_2$ to the monetary value of organic carbon, the $SSCO_2$ was multiplied by the ratio of total mass to carbon in a $CO_2$ molecule:

$$FP_{seq}(\$) = FPseq_{tot} \cdot SSCO_2 \cdot \frac{44}{12} \quad (5)$$

## Comparison to other blue carbon mechanisms
We compare the sequestered carbon from krill faecal pellets to coastal vegetation blue carbon stores. As krill faecal pellets are only one element of the contribution of krill to carbon sequestration, we also considered whether we could estimate the contribution from krill moults, carcasses and migrations, both daily and seasonally. We are confident in our ability to make the best estimates on the magnitude of moults and migrations. Firstly, Manno et al. [13] found that moult carbon contribution at 300 m equalled that of krill pellets in the Southern Ocean. We parsimoniously assume the sequestration by moults is equal to that of pellets. For migration, given that ~20 % of krill live deeper than 400 m[45], we estimate that the respired DIC released from these deep-dwelling krill, either from daily or seasonal migrations, is 26 MtC yr$^{-1}$. Please refer to the Supplementary Note S3 for more details. The values for salt marshes, seagrass and mangroves in terms of Mt C stored each year were taken from Bertram et al. [1]. Finally, the total biological carbon pump in our study region of the Southern Ocean was calculated using the OCIM model output[22], by taking POC flux at the sequestration depth ($FPT_{100}$) and summing over the area and time (October to April) of our sampling period.

## Tracing respired krill pellet carbon globally
We used the OCIM transport matrix to track carbon respired from krill pellets as they sink through the ocean. Summed krill pellet export fluxes (FPCexp) over the spring to early autumn period and the krill attenuation rate of $b = -0.30$ were used to find the steady-state distribution in the ocean of respired carbon originating from the pellets. Further, the global integral of this distribution as well as the average residence time (in years) was estimated. Note this does not include particulate carbon that escapes remineralisation and reaches the seafloor. This method has already been published in Boyd et al. [21] and Nowicki[32]. We also ran this analysis for a higher attenuation rate of $b = -0.86$ (Supplementary Fig. S4). The matrix equation is:

$$dC/dt = C\,(A-S) + Q = 0 \quad (6)$$

where A is the transport matrix, C(x,y,z) [gC m$^{-3}$] the concentration of dissolved inorganic carbon (DIC) within the ocean, Q(x, y, z) [gC m$^{-3}$ yr$^{-1}$] is the source of DIC (derived from observed surface pellet fluxes and a Martin curve), and S [yr$^{-1}$] represents the instantaneous removal of C from the surface. The steady-state solution: that is

$$C^* = -Q/(A-S) \quad (7)$$

is an estimate of the distribution of DIC emitted by Q in the world's oceans before it comes into contact with the atmosphere again. Integrating this over the entire ocean gives the total sequestered DIC. Specifically, writing V(x, y, z) [m3] as the volume of each grid box in the transport matrix representation, Cseq = ∑ C*V. Dividing this by the net rate at which DIC is injected into the ocean (i.e., the volume integral of Q) gives the mean residence time for DIC injected via Q below the surface.

## Reporting summary
Further information on research design is available in the Nature Portfolio Reporting Summary linked to this article.

## Data availability

The two forcing data sets used in this study can be accessed online. KRILLBASE density data can be downloaded from the British Antarctic Survey at https://www.bas.ac.uk/project/krillbase/#data. The OCIM model and data can be downloaded at https://tdevries.eri.ucsb.edu/models-and-data-products/.

## Code availability

The code used in this study for analysis and plotting is available at https://doi.org/10.6084/m9.figshare.26511022.

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

## Acknowledgements

E.L.C. was supported by an Imperial College Research Fellowship, a Natural Environment Research Council (NERC) grant NE/Y004515/1 and a WWF research grant (GB085708) along with A.A. and S.L.H. S.L.H. was additionally supported by the NERC British Antarctic Survey (BAS) ALI-Science Southern Ocean Ecosystems project. AA's contribution was also supported by the NERC PICCOLO programme. N.M. was funded by NERC grant NE/W001543/1. A.W.V. is supported by the Centre for Ocean Life, a VKR Centre of excellence funded by the Villum Foundation, and by the Horizon2020 projects ECOTIP (grant agreement #869383) and OCEAN-ICU (grant agreement #101083922). Figure 5 was produced by Visual Knowledge.

## Author contributions

E.L.C. devised the research and led the analysis and production of the figures and data presented in the main text. NM provided the carbon sequestration depth from the OCIM model. S.L.H. calculated the pellet egestion rate via the three different approaches. A.A. estimated the carbon sequestered via krill respiration. S.L.H. and A.A. provided expertise and advice on the circumpolar krill densities/abundance and the spatial area of krill habitat. A.B. and E.L.C. revised the attenuation rates. A.W.V. ran the transport matrix model to trace krill carbon in the global oceans. E.L.C. led the writing, and all authors contributed, with significant contributions from S.L.H.

## Competing interests

The authors declare no competing interests.
