## [Peer Review File · Nature Communications]

Antarctic krill sequester similar amounts of carbon to key coastal blue carbon habitatsREVIEWER COMMENTS

Reviewer #1 (Remarks to the Author):

This is a refreshing study that highlights the biological pump in the context of coastal blue carbon and the social cost of carbon.

I really like this study and the concept of Figure 1, but I have several concerns and suggestions about your definitions and labels:

- First, I think Fig 1B and 1C are mixed up in their labels in the caption and I would change the title of 1C from “Carbon sequestration depth” to something along the lines of “Minimum carbon sequestration depth”
- From my understanding of your Methods section, Fig 1b is FPCexp, which is actually a measure of krill fecal pellet production rate (egestion rate) per square meter, based on density of krill at ~20 m. I suggest changing the notation from “FPCexp” to “FPCprod” or “FPCegest” to stand for FP carbon production rate or egestion rate, not only in this graph but throughout the manuscript because this is not an export rate.
- Instead of showing this FP production map as Fig 1b, I think showing FPCseq (what I believe is Fig 1D) is more meaningful, but I would also change the notation for “FPCseq” here as well. Sequestration is usually in reference to an amount (mass) of carbon sequestered, not a rate. So what you are showing is the FPC export rate where Z is the depth at which the particles must surpass to be sequestered for over 100 years. I would change the label of “FPCseq” to “FPCflux”, because it seems that you define “FPCflux” in the Methods (Line 368; Equation 3), but don’t use that definition in the manuscript, and the only difference between “FPCflux” and “FPCseq” is that FPCflux uses a hypothetical mesopelagic Z, whereas FPCseq you define Z to be FPT100k, which is the main point and new contribution of this manuscript.
- In the final panel of Figure 1, I suggest showing a map of what you term FPseqtot, which I would suggest changing to “FPCseq”, for each corresponding map grid, where the units would be mgC/m².
- To summarize: Fig 1A is good. Move Fig 1C (depth needed to reach long term sequestration) to where Fig 1B is. Remove Fig 1B entirely since FP production is simply Fig 1A multiplied by a constant egestion factor. Move Fig 1D (“FPCflux”) to where 1C was. Add a map of the amount of FP carbon that can be sequestered for 100+ years within each grid cell you analyze (“FPCseq”) Please adjust the labels accordingly throughout the manuscript and tables and supplementary material.

Additionally, I believe the OCIM model is 2x2 degree spatial resolution, and it looks like you tested the krill FP export model at 2x2 degrees (along with several other resolutions), but chose to go with 2x6 degrees, why is that? Supplementary Figure 6 illustrates the differences, but why did you pick 2x6 for the krill and 2x2 for the circulation model? It would be helpful to have that explained in the Methods section.

Lines 95-97: Is this the OCIM model? You mention it here, but name it much later in Lines 247-248. Please briefly explain the key components of the ocean circulation model, such as is it global, is it an inverse model, what are the major parameters used in the model?

Lines 120-121: What do you mean by "plankton export rates"? Is this phytoplankton sinking? The export of all other zooplankton combined? I know there is a citation there, but this sentence should be more descriptive and explanatory.

Lines 247-248 and Figure 4: Please provide additional details, briefly. What is the timescale you are analyzing? This is a map of where krill FP carbon is after 1 year? Is this where krill FP carbon is found 100 years later?

Line 282: I think the first "suggest" is a typo and "is used to" should be changed to "can be used to".

Figure 5: Change the order of the rows:
areal extent (million km²) at the top
tC km⁻² yr⁻¹ next
MtC yr⁻¹ on bottom

Line 302: remove "does though"

Line 385-387: the sentence structure is confusing.

Line 429: change "the US were" to "the US was"

Lines 457 - 458: "used to find where respired carbon originating from the pellets ends up in the ocean" - over what timescale?

For the Supplementary Material:

Line 58: "2) daily mass krill both" - I think there is an incomplete sentence or typo in here

Table S3: What is SG, AP, and RS?

Table S4: Why are you putting the source/citation in twice?

DVM respiration analysis: are you assuming the krill that live below 400m are feeding at the surface and 1% of their biomass is egested as fecal pellets at depth >400m and 1% is converted into growth biomass? And then you are assuming that the 1% turned into growth will eventually die at depth (as opposed to dying in surface waters) and sink and be sequestered?

Does this take into account that they may be feeding and respiring at depth, which leads to attenuation of their own/overall carbon export?

Line 298: "and if 4% is available then 2% remains..." This part of the sentence is very confusing. 4%

of what is available for what?

Reviewer #2 (Remarks to the Author):

**Review of NCOMMS-23-46280-T by Cavan et al.
Valuing carbon sequestration by Antarctic krill faecal pellets**

In this study, Cavan et al. provide an estimate on the circumpolar export and sequestration of krill faecal pellet carbon. For this, the authors build on existing krill abundance data from KRILLBASE, krill faecal pellet fluxes from Belcher et al. 2017 and Pauli et al. 2021, as well as on an ocean circulation model. The study finds that in austral spring and autumn, krill faecal pellets sequester about 20 MtC, which is in the same order as other blue carbon stores. Moreover, the study provides an estimate of the economic value of the ecological service krill faecal pellets provide by sequestering carbon for > 100 yrs, equating to USD \$4-46. Additional calculations on sequestration depths and the trajectory of krill pellet derived DIC to other ocean regions make this manuscript a valuable addition to the literature. If published, the manuscript will join a series of articles published in Nature Communications over the past few years highlighting the great importance of Antarctic krill for the global carbon sink (Belcher et al 2019, Cavan et al. 2019, Manno et al. 2020, Pauli et al. 2021).

The manuscript is well written, comprehensible and the most recent and relevant literature is cited. The analyses and calculations seem robust and valid, limitations and choices for single parameters are carefully explained and discussed; sensitivity and uncertainty analyses were conducted and the results are discussed in the context of other published studies.

I suggest the following revisions.

Main

General:

The ms does not follow the traditional structure of introduction, results, discussion and methods, and as this is not a classical field or laboratory study, that's fine. However, particularly in the first half of the ms, I find the current structure a bit confusing; the first subheading in line 90 does not seem to properly reflect what the respective section deals with. The section is entitled "circumpolar krill faecal pellet flux" but mainly deals with sequestration and the description of figure 1. Following the introductory part (lines 45-88) a few sentences on the methods used would help the reader to better understand the following analyses and figures. At the end of the main part, an additional subheading indicating the concluding paragraphs would be nice (see my comment below).

Specific comments:

Line 53: 'unconsumed' - consumption is not the only factor of attenuation, as you've mentioned in other parts of this ms, e.g. line 79. Suggest to rephrase and to include remineralisation here.

Line 59: I think it would be helpful to include a one-time definition of austral spring and autumn (i.e. which months).

Lines 59-63: This is a very long sentence. Suggest making it two.

Line 70: Potentially for the discussion part, do you have any suggestions on which other pelagic organisms this could/should be applied to?

Line 73: I suggest to mention Atkinson et al. by name and/or KRILLBASE. At least to readers who are broadly familiar with the topic this will immediately give an idea of the dataset you worked with. Given the ongoing debate on the southward-retreat of krill and declining abundances, I further suggest to mention the time period the abundance data from Atkinson et al are based on.

Figure 1: Suggest to remove 'January' from the figure title, as this does not apply for 1c (as described below). A description similar to lines 92-93 could be added here for clarity. Numbers and letters on left and right hand side of the figure are cropped. This applies to all figures of this type (Suppl. Fig. S1, Suppl. Fig. S2a, Suppl. Fig. S7).

Lines 102-105. That's a very interesting finding and would indicate that results in previous studies such as Manno et al. 2020 and Pauli et al. 2021, who reported export fluxes to 300m, have potentially shown real sequestered carbon fluxes.

Line 111/Table 1: Suggest to rephrase "across the entire *spatial dataset*" or something along those lines. I personally find the references to Figs. 1 and 2 a bit confusing.

Line 145: I don't quite understand why April is mentioned separately in brackets. Is it not included in the spring/autumn months elsewhere? See my comment above to include a definition at the beginning of the ms.

Figure 2: If I understand correctly, the main or only difference in the data presented here compared to Table 1 is the fixed value for USD \$ per t CO₂? If needed, this figure could be omitted/moved to the supplementary material, or maybe it could be combined with table 1 somehow?

Line 211: Suggest to repeat parameters here.

Line 212. Open bracket

Line 241/Figure 3. Suggest consistent use of terms in figure and caption (krill density vs. abundance).

Line 271. There is a discrepancy with fig 5. In the figure, the global blue carbon store of seagrass is 50 MtC yr⁻¹, in the text here it says 44.

Line 282. Typo in 'suggests we suggest'

Line 283/Figure 5: I am not sure what is meant by "season or year". Do the carbon stores presented relate to different time periods? If so, this needs to be clarified in the figure. I am also not sure what the arrows 'seasonal' and 'daily' mean.

Lines 285-310. Suggest to give this section a subheading, entitled 'conclusion' or similar. Lines 288-290 need referencing.

Methods

Lines 333-338: I understand the reasoning given here to cap the krill abundance data to 600 Ind. m⁻². However, vast krill swarms and resulting punctual very high pellet export fluxes (as reported e.g. by Pauli et al. 2021 and elsewhere) do play a role and influence the circumpolar and annual mean fluxes. The sensitivity analysis conducted here also shows that abundance has the largest effect on overall carbon sequestration. Thus, it would be good to include a few more discussing sentences here or, if possible, even include a small calculation/estimate on how the results presented here could change if higher abundances were included.

Line 387: Do you have an estimate for the magnitude of this potential in-or decrease?

Reviewer #3 (Remarks to the Author):

The submission by Cavan et al estimates the amount carbon exported by Antarctic krill that reaches deep water masses where the respired carbon will be sequestered longer than 100 years, and converts this into a dollar "value" for comparison to other "blue carbon" reservoirs. They conclude that the carbon sequestration "service" provided by krill is at least as valuable as that provided by mangrove ecosystems, and therefore warrant similar attention from a conservation standpoint. I find this to be a refreshing and interesting study – to my knowledge this approach has not been applied to open-ocean biological carbon pump processes before, and this study will help move those processes into the conversation about blue carbon and ecosystem services, which has previously been dominated by coastal marine ecosystems. The analysis in this paper is quite rigorous, and while the results are subject to a wide range of uncertainties, the study does a good job of acknowledging those uncertainties and highlighting where future work is needed to better "value" biological pump processes. Overall, I am supportive of this paper and hope to see it published in Nature Communications upon revision. There are only two places where I would recommend additional work to improve the paper, both regarding comparison to previous datasets. I will expand on these below, followed by a set of minor comments about the writing.

First, to assess their estimates of faecal pellet production, the authors compare to a satellite-based

estimate of integrated Southern Ocean NPP from Arrigo et al., 2008. I recommend that the authors modify this to compare to some more up-to-date NPP estimates that employ newer satellite sensors and updated algorithms. I assume the Arrigo estimate was chosen because it uses an algorithm especially calibrated for the Southern Ocean, but a recent assessment (Arteaga et al., 2018) showed that a range of more recent satellite estimates (VGPM, CbPM, Marra) can reproduce Southern Ocean observations just as successfully, and tend to find higher NPP polewards of 50S than the Arrigo estimate (average 2.7Gt/yr, rather than 1.9Gt/yr in Arrigo). A second advantage of using these newer estimates is that the authors could obtain the spatially-resolved data and do a more granular comparison than just the Southern Ocean integral. Specifically, the authors find that the majority of krill pellet production is focused in the Atlantic sector of the Southern Ocean – it would be a very valuable reference point to know what fraction of NPP in that specific region would need to be routed to krill pellets in order to explain the model predictions.

Second, the authors compare their estimates of krill pellet export to the predictions of a previous food web model by DeVries and Weber, 2017. There has recently been an update to this model (Nowicki et al., 2022), which separates export into faecal pellets and phytoplankton aggregates and estimates the depth penetration of both components. The full output of the model is provided through the supplement, and would allow for a more apples-to-apples comparison. (NOTE: On second reading I saw that the Nowicki paper is included in the reference list but I did not see it cited in this comparison, which is perhaps a simple omission). The authors should also be warned that in comparing to these models (either Weber & Devries or Nowicki), their krill pellet export estimates should probably not be considered as “additional to” (line 166) the model export estimates. In those models, total carbon export is constrained by tuning the model to match a range of tracer observations, including oxygen, POC, and DOC, and it is therefore unlikely true export could be 60% higher. Instead, the model has likely folded in the krill export estimates into one of the processes it does resolve (particle settling, or copepod pellets in the case of Nowicki et al.), and therefore this is a case of mis-identified carbon export, rather than missing carbon export. Note, the mis-identification is still important and worth noting, insofar as krill faecal pellets seem to behave differently from other particles in terms remineralization depth.

With these two points of comparison revised, I would have further objections to this very interesting paper being published!

Minor points:

Line 29: Change “The vast area of ocean krill inhabit...” to “The vast area that ocean krill inhabit...” or “The vast area of ocean krill habitat...”

Line 59: Consider splitting sentence beginning “Krill face...”. This is a long and somewhat confusing sentence.

Line 164: Who is “their” in “their model”? Due to the reference style, no authors have been mentioned.

Line 188: “assess whether”

Line 244: Revise to “Carbon transferred to the deep ocean by Antarctic krill...”

Line 348: There is an unclosed parenthesis in this sentence.

Fig. 1 Caption: Panels b and c are referred to the wrong way round.

Arteaga, L., N. Haentjens, E. Boss, K.S. Johnson, and J.L. Sarmiento. 2018. Assessment of Export Efficiency Equations in the Southern Ocean Applied to Satellite-Based Net Primary Production. *Journal of Geophysical Research-Oceans* 123(4):2945-2964, 10.1002/2018jc013787.

Nowicki, M., T. DeVries, and D.A. Siegel. 2022. Quantifying the Carbon Export and Sequestration Pathways of the Ocean's Biological Carbon Pump. *Global Biogeochemical Cycles* 36(3), ARTN e2021GB007083
10.1029/2021GB007083.

Response to reviewers

Re. Valuing carbon sequestration by Antarctic krill faecal pellets, Cavan et. al.

We thank all three reviewers for their considered feedback and respond to each point in blue below. We would like to let all reviewers know that we revised the above title, now to read:

‘Antarctic krill sequester similar total amounts of carbon to key coastal blue carbon habitats’.

We changed the title as the key focus of the manuscript, as highlighted and welcomed by the reviewers, is placing the krill carbon sequestration amount (mass per unit of time) in the context of more common blue carbon reservoirs, and to highlight the conservation need for Antarctic krill. We believe this new title reflects the purpose of our study better.

Reviewer #1 (Remarks to the Author):

This is a refreshing study that highlights the biological pump in the context of coastal blue carbon and the social cost of carbon.

I really like this study and the concept of Figure 1, but I have several concerns and suggestions about your definitions and labels:

- First, I think Fig 1B and 1C are mixed up in their labels in the caption and I would change the title of 1C from “Carbon sequestration depth” to something along the lines of “Minimum carbon sequestration depth”
- From my understanding of your Methods section, Fig 1b is FPCexp, which is actually a measure of krill fecal pellet production rate (egestion rate) per square meter, based on density of krill at ~20 m. I suggest changing the notation from “FPCexp” to “FPCprod” or “FPCegest” to stand for FP carbon production rate or egestion rate, not only in this graph but throughout the manuscript because this is not an export rate.
- Instead of showing this FP production map as Fig 1b, I think showing FPCseq (what I believe is Fig 1D) is more meaningful, but I would also change the notation for “FPCseq” here as well. Sequestration is usually in reference to an amount (mass) of carbon sequestered, not a rate. So what you are showing is the FPC export rate where Z is the depth at which the particles must surpass to be sequestered for over 100 years. I would change the label of “FPCseq” to “FPCflux”, because it seems that you define “FPCflux” in the Methods (Line 368; Equation 3), but don’t use that definition in the manuscript, and the only difference between “FPCflux” and “FPCseq” is that FPCflux uses a hypothetical mesopelagic Z, whereas FPCseq you define Z to be FPT100k, which is the main point and new contribution of this manuscript.
- In the final panel of Figure 1, I suggest showing a map of what you term FPseqtot, which I would suggest changing to “FPCseq”, for each corresponding map grid, where the units would be mgC/m².
- To summarize: Fig 1A is good. Move Fig 1C (depth needed to reach long term sequestration) to where Fig 1B is. Remove Fig 1B entirely since FP production is simply Fig 1A multiplied by a constant egestion factor. Move Fig 1D (“FPCflux”) to where 1C was. Add a map of the amount of FP carbon that can be sequestered for 100+ years within each grid cell you analyze (“FPCseq”) Please adjust the labels accordingly throughout the manuscript and tables and supplementary material.

Thank you for taking the time to carefully and thoughtfully review our work. We are grateful for your feedback which has helped the readability of the manuscript, particularly the point above regarding Figure 1. We rearranged the figures as you suggested. We plotted the FPseqtot but decided not to include it in the final version because it was just ‘FPCflux’ at sequestration depth multiplied by time which is constant, so spatially does not give any new information. In addition, a similar figure exists

as Fig. 4 which does this in more detail using the transport matrix model and at steady state. It continues to show the Atlantic as the high carbon sequestration regions (in units of mass per m²). The new Fig.1 includes krill density (a), sequestration depth (b) and FP flux at sequestration depth (c).

Additionally, I believe the OCIM model is 2x2 degree spatial resolution, and it looks like you tested the krill FP export model at 2x2 degrees (along with several other resolutions), but chose to go with 2x6 degrees, why is that? Supplementary Figure 6 illustrates the differences, but why did you pick 2x6 for the krill and 2x2 for the circulation model? It would be helpful to have that explained in the Methods section.

The reason to increase the cell size was to cover a large surface area of the Southern Ocean so we could scale to be nearly circumpolar/cover the entire area. We have explained this further in line 370.

Lines 95-97: Is this the OCIM model? You mention it here, but name it much later in Lines 247-248. Please briefly explain the key components of the ocean circulation model, such as is it global, is it an inverse model, what are the major parameters used in the model?

Yes this is correct, we have provided further explanation.

Lines 120-121: What do you mean by "plankton export rates"? Is this phytoplankton sinking? The export of all other zooplankton combined? I know there is a citation there, but this sentence should be more descriptive and explanatory.

We have defined this better.

Lines 247-248 and Figure 4: Please provide additional details, briefly. What is the timescale you are analyzing? This is a map of where krill FP carbon is after 1 year? Is this where krill FP carbon is found 100 years later?

This is at steady state in the ocean, and we have described this better in lines 278+.

Line 282: I think the first "suggest" is a typo and "is used to" should be changed to "can be used to".

Yes, thank you for finding this.

Figure 5: Change the order of the rows:
areal extent (million km²) at the top
tC km⁻² yr⁻¹ next
MtC yr⁻¹ on bottom

This is a good idea and we have changed the order.

Line 302: remove "does though" Done

Line 385-387: the sentence structure is confusing. Agreed so we have edited this.

Line 429: change "the US were" to "the US was" Thank you, we changed this.

Lines 457 - 458: "used to find where respired carbon originating from the pellets ends up in the ocean" - over what timescale? See explanation given regarding Fig. 4 and lines 278+.

For the Supplementary Material:

Line 58: "2) daily mass krill both" - I think there is an incomplete sentence or typo in here Yes thank you, this is now edited.

Table S3: What is SG, AP, and RS? South Georgia, Antarctic Peninsula and Ross Sea, we have explained in full now in text and apologise it was not originally clear.

Table S4: Why are you putting the source/citation in twice? Edited

DVM respiration analysis: are you assuming the krill that live below 400m are feeding at the surface and 1% of their biomass is egested as fecal pellets at depth >400m and 1% is converted into growth biomass? And then you are assuming that the 1% turned into growth will eventually die at depth (as opposed to dying in surface waters) and sink and be sequestered?

We realise now that this whole section was slightly badly written and have edited the whole section now to explain how we estimate the respiratory flux. Our estimate of 26 MtC per year is not based on krill dying but instead only on their respiration at depth.

Does this take into account that they may be feeding and respiring at depth, which leads to attenuation of their own/overall carbon export?

This calculation assumes for simplicity that the carbon respired at depths greater than 400 m was eaten in the more productive surface layers. We have added a paragraph in this section in the supplementary materials to explain this assumption more clearly and that it may not completely hold, as well as a series of three other considerations which would increase our estimate of C flux due to krill migration. Importantly we emphasise (lines 443+ in the supplementary material) the uncertainties in all of these calculations.

Line 298: "and if 4% is available then 2% remains..." This part of the sentence is very confusing. 4% of what is available for what?

We agree this section was unclear and we have now explained more clearly the partitioning of summer carbon rations of krill where the daily C ingestion rate of krill equates to ~4% of their body C per day, and this is partitioned roughly into 1% for growth, 2% for egestion of unassimilated food and the remaining 2% is respired. This is now explained in lines 434+ in the supplementary material.

Reviewer #2 (Remarks to the Author):

**Review of NCOMMS-23-46280-T by Cavan et al.
Valuing carbon sequestration by Antarctic krill faecal pellets**

In this study, Cavan et al. provide an estimate on the circumpolar export and sequestration of krill faecal pellet carbon. For this, the authors build on existing krill abundance data from KRILLBASE, krill faecal pellet fluxes from Belcher et al. 2017 and Pauli et al. 2021, as well as on an ocean circulation model. The study finds that in austral spring and autumn, krill faecal pellets sequester about 20 MtC, which is in the same order as other blue carbon stores. Moreover, the study provides an estimate of the economic value of the ecological service krill faecal pellets provide by sequestering carbon for > 100 yrs, equating to USD \$4-46. Additional calculations on sequestration depths and the trajectory of krill pellet derived DIC to other ocean regions make this manuscript a valuable addition to the

literature. If published, the manuscript will join a series of articles published in Nature Communications over the past few years highlighting the great importance of Antarctic krill for the global carbon sink (Belcher et al 2019, Cavan et al. 2019, Manno et al. 2020, Pauli et al. 2021).

The manuscript is well written, comprehensible and the most recent and relevant literature is cited. The analyses and calculations seem robust and valid, limitations and choices for single parameters are carefully explained and discussed; sensitivity and uncertainty analyses were conducted and the results are discussed in the context of other published studies.

I suggest the following revisions.

Thank you for your considered and positive assessment of this manuscript. We have followed all your suggestions as detailed below.

Main

General:

The ms does not follow the traditional structure of introduction, results, discussion and methods, and as this is not a classical field or laboratory study, that's fine. However, particularly in the first half of the ms, I find the current structure a bit confusing; the first subheading in line 90 does not seem to properly reflect what the respective section deals with. The section is entitled "circumpolar krill faecal pellet flux" but mainly deals with sequestration and the description of figure 1. Following the introductory part (lines 45-88) a few sentences on the methods used would help the reader to better understand the following analyses and figures. At the end of the main part, an additional subheading indicating the concluding paragraphs would be nice (see my comment below).

We have edited the first title and expanded on the methods paragraph at the end of the introduction. We agree that our initial attempt to describe the study at the end of the introduction was insufficient. We have also added a sentence to summarise our main finding at the end of the Introduction, in accordance to the style requirements of the journal.

Specific comments:

Line 53: 'unconsumed' - consumption is not the only factor of attenuation, as you've mentioned in other parts of this ms, e.g. line 79. Suggest to rephrase and to include remineralisation here. Agree, edited.

Line 59: I think it would be helpful to include a one-time definition of austral spring and autumn (i.e. which months). Agreed so we have now edited.

Lines 59-63: This is a very long sentence. Suggest making it two. Done.

Line 70: Potentially for the discussion part, do you have any suggestions on which other pelagic organisms this could/should be applied to? Added some suggestions in lines 317.

Line 73: I suggest to mention Atkinson et al. by name and/or KRILLBASE. At least to readers who are broadly familiar with the topic this will immediately give an idea of the dataset you worked with. Given the ongoing debate on the southward-retreat of krill and declining abundances, I further suggest to mention the time period the abundance data from Atkinson et al are based on. This is a good idea so we edited the text here.

Figure 1: Suggest to remove 'January' from the figure title, as this does not apply for 1c (as described below). A description similar to lines 92-93 could be added here for clarity. Numbers and letters on left and right hand side of the figure are cropped. This applies to all figures of this type (Suppl. Fig.

S1, Suppl. Fig. S2a, Suppl. Fig. S7). Done, and thank you for highlighting about the cropping, I think this was an issue when converting to PDF.

Lines 102-105. That's a very interesting finding and would indicate that results in previous studies such as Manno et al. 2020 and Pauli et al. 2021, who reported export fluxes to 300m, have potentially shown real sequestered carbon fluxes. This is a good point which we have bought into the discussion here. We have now also emphasised the importance of sometimes shallow depths needed for sequestration in the Abstract (line 29).

Line 111/Table 1: Suggest to rephrase "across the entire *spatial dataset*" or something along those lines. I personally find the references to Figs. 1 and 2 a bit confusing. Done.

Line 145: I don't quite understand why April is mentioned separately in brackets. Is it not included in the spring/autumn months elsewhere? See my comment above to include a definition at the beginning of the ms. We have rephrased this sentence so April is no longer in brackets. It was because April is Autumn, but krill abundance is still moderate we included data for April in the data analysis instead of just 'spring' and 'summer' months. We have edited the text to explain this better.

Figure 2: If I understand correctly, the main or only difference in the data presented here compared to Table 1 is the fixed value for USD \$ per t CO₂? If needed, this figure could be omitted/moved to the supplementary material, or maybe it could be combined with table 1 somehow? Correct but we chose to leave it in as a simplest representation of our results, that the Atlantic Ocean dominates carbon sequestration by krill.

Line 211: Suggest to repeat parameters here. Done.

Line 212. Open bracket Noted, thank you.

Line 241/Figure 3. Suggest consistent use of terms in figure and caption (krill density vs. abundance). Changed to krill density throughout.

Line 271. There is a discrepancy with fig 5. In the figure, the global blue carbon store of seagrass is 50 MtC yr⁻¹, in the text here it says 44. Thank you for spotting this! The figure has now been corrected.

Line 282. Typo in 'suggests we suggest' Edited.

Line 283/Figure 5: I am not sure what is meant by "season or year". Do the carbon stores presented relate to different time periods? If so, this needs to be clarified in the figure. I am also not sure what the arrows 'seasonal' and 'daily' mean. This has been simplified and edited in the figure caption.

Lines 285-310. Suggest to give this section a subheading, entitled 'conclusion' or similar. Lines 288-290 need referencing. Done.

Methods

Lines 333-338: I understand the reasoning given here to cap the krill abundance data to 600 Ind. m⁻². However, vast krill swarms and resulting punctual very high pellet export fluxes (as reported e.g. by Pauli et al. 2021 and elsewhere) do play a role and influence the circumpolar and annual mean fluxes. The sensitivity analysis conducted here also shows that abundance has the largest effect on

overall carbon sequestration. Thus, it would be good to include a few more discussing sentences here or, if possible, even include a small calculation/estimate on how the results presented here could change if higher abundances were included.

We calculated that removing these 7 data points reduced the total sequestered from 21.5 MtC yr⁻¹ to 20MtC yr⁻¹. The data were from one net haul, i.e. one observation, with the 7 data points coming from extrapolating the observations over time (n = 7 months). We have included this estimate of sequestration in the Methods section.

Line 387: Do you have an estimate for the magnitude of this potential in-or decrease? This would be really interesting to know, however we have included some text here stating that the model does not exist in finer resolution so we cannot test it at present.

Reviewer #3 (Remarks to the Author):

The submission by Cavan et al estimates the amount carbon exported by Antarctic krill that reaches deep water masses where the respired carbon will be sequestered longer than 100 years, and converts this into a dollar “value” for comparison to other “blue carbon” reservoirs. They conclude that the carbon sequestration “service” provided by krill is at least as valuable as that provided by mangrove ecosystems, and therefore warrant similar attention from a conservation standpoint. I find this to be a refreshing and interesting study – to my knowledge this approach has not been applied to open-ocean biological carbon pump processes before, and this study will help move those processes into the conversation about blue carbon and ecosystem services, which has previously been dominated by coastal marine ecosystems. The analysis in this paper is quite rigorous, and while the results are subject to a wide range of uncertainties, the study does a good job of acknowledging those uncertainties and highlighting where future work is needed to better “value” biological pump processes. Overall, I am supportive of this paper and hope to see it published in Nature Communications upon revision. There are only two places where I would recommend additional work to improve the paper, both regarding comparison to previous datasets. I will expand on these below, followed by a set of minor comments about the writing.

First, to assess their estimates of faecal pellet production, the authors compare to a satellite-based estimate of integrated Southern Ocean NPP from Arrigo et al., 2008. I recommend that the authors modify this to compare to some more up-to-date NPP estimates that employ newer satellite sensors and updated algorithms. I assume the Arrigo estimate was chosen because it uses an algorithm especially calibrated for the Southern Ocean, but a recent assessment (Arteaga et al., 2018) showed that a range of more recent satellite estimates (VGPM, CbPM, Marra) can reproduce Southern Ocean observations just as successfully, and tend to find higher NPP polewards of 50S than the Arrigo estimate (average 2.7Gt/yr, rather than 1.9Gt/yr in Arrigo). A second advantage of using these newer estimates is that the authors could obtain the spatially-resolved data and do a more granular comparison than just the Southern Ocean integral. Specifically, the authors find that the majority of krill pellet production is focused in the Atlantic sector of the Southern Ocean – it would be a very valuable reference point to know what fraction of NPP in that specific region would need to be routed to krill pellets in order to explain the model predictions.

Thank you for reviewing our work and for your suggestions of additional analyses and comparisons which add great value.

The Arteaga et al. 2018 value of NPP south of 50S (our study region) is in fact 2 GtC yr⁻¹. From a previous collaboration with the authors we actually have that data and were able to analyse it to calculate this number. In our study, we used the Arrigo value which was almost identical, at 1.9 GtC

yr⁻¹ for NPP. Therefore the Arrigo value holds as within a valid range of NPP and we continue to use this value.

The 2.7 GtC yr⁻¹ Arteaga report actually refers to the total export of carbon in their study region (30S), rather than NPP. Their NPP value is ~16Gt C yr⁻¹ because of the larger study area (i.e. including 30-50S) and the many landmasses and subsequent iron fertilised coastal regions with high NPP in the more equatorward latitudes.

We ran the additional analysis suggested which was a really interesting find, expressed in an additional supplementary table (Table S6). We didn't include more figures as spatially the result was similar/the same to Fig. S2a, but with different units/scales on the z-scale (colours). Overall, 2.5 % of NPP in our study area ends up as krill pellet carbon that is sequestered, but this goes up to 74 % in krill swarms. See lines 175+.

Second, the authors compare their estimates of krill pellet export to the predictions of a previous food web model by DeVries and Weber, 2017. There has recently been an update to this model (Nowicki et al., 2022), which separates export into faecal pellets and phytoplankton aggregates and estimates the depth penetration of both components. The full output of the model is provided through the supplement, and would allow for a more apples-to-apples comparison. (NOTE: On second reading I saw that the Nowicki paper is included in the reference list but I did not see it cited in this comparison, which is perhaps a simple omission). The authors should also be warned that in comparing to these models (either Weber & Devries or Nowicki), their krill pellet export estimates should probably not be considered as "additional to" (line 166) the model export estimates. In those models, total carbon export is constrained by tuning the model to match a range of tracer observations, including oxygen, POC, and DOC, and it is therefore unlikely true export could be 60% higher. Instead, the model has likely folded in the krill export estimates into one of the processes it does resolve (particle settling, or copepod pellets in the case of Nowicki et al.), and therefore this is a case of mis-identified carbon export, rather than missing carbon export. Note, the mis-identification is still important and worth noting, insofar as krill faecal pellets seem to behave differently from other particles in terms remineralization depth.

Thank you for highlighting how this carbon should be interpreted when compared to the BCP, this is a very important point! We have now re-written the text around this to suggest most wouldn't be additional. We also downloaded the Nowicki data and replotted Fig.S2a using this data.

With these two points of comparison revised, I would have further objections to this very interesting paper being published!

Minor points:

Line 29: Change "The vast area of ocean krill inhabit..." to "The vast area that ocean krill inhabit..." or "The vast area of ocean krill habitat..." Thank you, now edited.

Line 59: Consider splitting sentence beginning "Krill face...". This is a long and somewhat confusing sentence. Thank you, edited.

Line 164: Who is "their" in "their model"? Due to the reference style, no authors have been mentioned. Edited.

Line 188: "assess whether" Edited.

Line 244: Revise to "Carbon transferred to the deep ocean by Antarctic krill..." Thank you, edited.

Line 348: There is an unclosed parenthesis in this sentence. Edited

Fig. 1 Caption: Panels b and c are referred to the wrong way round. Edited.

Arteaga, L., N. Haentjens, E. Boss, K.S. Johnson, and J.L. Sarmiento. 2018. Assessment of Export Efficiency Equations in the Southern Ocean Applied to Satellite-Based Net Primary Production. *Journal of Geophysical Research-Oceans* 123(4):2945-2964, [10.1002/2018jc013787](https://doi.org/10.1002/2018jc013787).

Nowicki, M., T. DeVries, and D.A. Siegel. 2022. Quantifying the Carbon Export and Sequestration Pathways of the Ocean's Biological Carbon Pump. *Global Biogeochemical Cycles* 36(3), ARTN e2021GB007083
[10.1029/2021GB007083](https://doi.org/10.1029/2021GB007083).

REVIEWERS' COMMENTS

Reviewer #1 (Remarks to the Author):

The manuscript is much improved. Thank you for taking the time to enhance the manuscript for publication. The authors have addressed all of my questions and concerns.

There are some minor formatting issues in the Supplemental Materials with regards to the line numbers and tables - I'm not sure if that is something the Authors have to fix or the type-editors of Nat Comms will fix.

Reviewer #3 (Remarks to the Author):

I think that the authors have done an admirable job of addressing my own concerns, and those of two additional reviewers.

I'm glad that the authors acted on my suggestion to make a spatially-resolved comparison to satellite NPP data, and given that this did not significantly change their conclusions I'm happy with the approach of tucking the result into a supplementary table. I also find the discussion related to the food-web model comparison much improved - the authors now point out that the krill carbon export would have been mis-allocated in those previous models, rather than omitted altogether.

On second read, I find the paper just as novel and interesting as I did last time round, now with previous weaknesses shored up through review. In my opinion it is ready for publication, and I look forward to seeing it in Nature Communications, and assigning it as reading in class!

If I could make one final (and very minor) nitpick, it would be that Figure 2 should have a horizontal zero line added, or have the y-axis redefined to start at zero. It looks a little awkward with the bars floating above the x axis.

Congratulations to the authors on a very nice paper,
Tom Weber